# Immunohistochemical Evaluation of Candidate Biomarkers for Fluorescence-Guided Surgery of Myxofibrosarcoma Using an Objective Scoring Method

**DOI:** 10.3390/biomedicines11030982

**Published:** 2023-03-22

**Authors:** Zeger Rijs, Esther Belt, Gijsbert M. Kalisvaart, Cornelis F. M. Sier, Peter J. K. Kuppen, Arjen H. G. Cleven, Alexander L. Vahrmeijer, Michiel A. J. van de Sande, Pieter B. A. A. van Driel

**Affiliations:** 1Department of Orthopedic Surgery, Leiden University Medical Center, 2333 ZA Leiden, The Netherlands; 2Department of Radiology, Leiden University Medical Center, 2333 ZA Leiden, The Netherlands; 3Department of Surgery, Leiden University Medical Center, 2333 ZA Leiden, The Netherlands; 4Percuros BV, Zernikedreef 8, 2333 CL Leiden, The Netherlands; 5Department of Pathology, Leiden University Medical Center, 2333 ZA Leiden, The Netherlands; 6Department of Pathology, University Medical Center Groningen, 9700 RB Groningen, The Netherlands; 7Department of Orthopedic Surgery, Isala Hospital, 8025 AB Zwolle, The Netherlands

**Keywords:** soft tissue sarcoma, molecular imaging, immunohistochemistry, fluorescence-guided surgery, tumor endothelial marker-1

## Abstract

Introduction: Myxofibrosarcoma (MFS) is the most common soft-tissue sarcoma subtype in elderly patients. Local recurrence (LR) remains a major concern as the lack of intraoperative guidance and an infiltrative growth pattern with long, slender tails hamper surgeons’ ability to achieve adequate resection margins for MFS. Fluorescence-guided surgery (FGS) could overcome this concern by delineating tumor tissue during surgery. One of the most important steps to successful FGS is to define a tumor-specific biomarker that is highly overexpressed in tumor tissue while low or absent in adjacent healthy tissue. The aim of this study is to evaluate the expression of eight previously selected promising biomarkers for FGS in MFS tissue samples with adjacent healthy tissue using immunohistochemistry (IHC). Methods: The following eight biomarkers were stained in seventeen paraffin-embedded MFS samples: tumor endothelial marker-1 (TEM-1, also known as endosialin/CD248), vascular endothelial growth factor receptor-1 (VEGFR-1, also known as Flt-1), vascular endothelial growth factor receptor-2 (VEGFR-2, also known as Flk1), vascular endothelial growth factor-A (VEGF-A), epidermal growth factor receptor (EGFR), insulin-like growth factor-1 receptor (IGF-1R), platelet derived growth factor receptor-α (PDGFR-α), and cluster of differentiation 40 (CD40, also known as TNFRSF5). A pathologist specializing in sarcoma annotated the margin between the tumor and adjacent healthy tissue in each MFS tissue sample. Subsequently, we used an objective IHC scoring method to assess and compare the difference in staining intensity between the tumor and adjacent healthy tissue, which is crucial for the use of FGS. Results: TEM-1, VEGF-A, and PDGFR-α stained all MFS tumors, while the other biomarkers did not show expression in all MFS tumors. Ultimately, TEM-1 was identified as the most suitable biomarker for FGS in MFS based on higher tumor-to-background (TBR) staining intensity compared to VEGF-A and PDGFR-α, regardless of preoperative therapy. Conclusion: TEM-1-targeted FGS tracers should be further investigated to optimize MFS treatment.

## 1. Introduction

Myxofibrosarcoma (MFS) is the most common soft tissue sarcoma (STS) subtype in elderly patients, with a peak incidence in 60–69-year-old patients [1,2]. To optimize treatment, MFS patients are referred to specialized sarcoma centers, where treatment is discussed in multidisciplinary tumor boards before surgery [3,4]. Despite increased complication risks associated with radiotherapy, localized MFS is currently treated with (neo)adjuvant radiotherapy combined with often extensive surgery [5,6]. This is because local recurrence (LR) is a major concern, occurring in 28–40% of MFS cases within a 5 year follow-up period and often associated with inadequate surgical margins [7,8]. Inherently, surgical margins are a significant predictor of survival [9]. Yet, sarcoma surgeons still rely on preoperative imaging modalities, such as MRI and CT, to identify the exact tumor location and margins. Translation of these preoperative images to the operating theatre is challenging, as surgeons only have their hands and eyes to rely on during surgery. In addition, the position of soft tissue changes during surgery, and MFS has an infiltrative growth pattern with long, slender tails that are difficult to detect with preoperative imaging [10]. Consequently, the surgical margins are difficult to appreciate and manage during surgery [11]. Efforts have been made to unravel the pathogenesis and identify druggable targets with molecular analyses, and targeting the cell cycle in these tumors could be a powerful approach [12,13]. However, this has not yet led to improved outcomes; a recent cohort study including 908 MFS patients, diagnosed between 2002 and 2019, highlighted that LR rates (39%) and survival (five year overall survival of 68%) did not improve during the study period, emphasizing the need to improve treatment strategies [14].

An upcoming tool and a different approach to achieving adequate margins in surgical oncology is fluorescence-guided surgery (FGS), which is based on near-infrared fluorescence (NIRF). FGS real-time intraoperative imaging helps the surgeon to discriminate tumors from adjacent healthy tissue. It can be subdivided into non-targeted FGS and targeted (tumor-specific) FGS. Non-targeted FGS mostly uses the fluorescent dye indocyanine green (ICG). Once ICG is administered, it accumulates in tumors due to their leaky vascular capillaries, which is referred to as the enhanced permeability and retention (EPR) effect [15]. However, this EPR effect is known to be influenced by many factors, such as tumor type, size, presence of necrosis, location, inflammation, and vascular mediators, as has been demonstrated for sarcoma surgery [16]. Therefore, the signal intensity of ICG is unpredictable, and will vary greatly between heterogeneous sarcoma tumors. False negativity could occur in cases with very small nodules, nodules with extensive necrosis, or minimally viable tissue. False positivity could occur as well, for example, in tissues with reactive changes or high levels of vascular permeability mediators such as bradykinin and prostaglandin. This could lead to over-resection, resulting in more wound complications and increased functional impairments. Targeted FGS, which can be achieved by binding fluorophores to sarcoma-specific targeting moieties such as antibodies or peptides, could theoretically overcome these issues. Targeted FGS has already been explored for various tumor types with promising results and has the potential to play an important role in future sarcoma surgery [17,18,19].

Three important parameters define successful targeted FGS: the tumor-specific biomarker, the targeting moiety conjugated to a fluorophore (tracer), and the dedicated FGS camera system. While the ideal tumor-specific biomarker is strongly and diffusely expressed on tumor cells and absent on adjacent healthy tissue, a minimum of 1.5× overexpression of the tumor-specific biomarker on tumor cells compared to adjacent healthy tissue is already enough to guide surgeons in their actions [20]. The biomarker should preferably be located on the cell surface of malignant cells to permit direct targeting and have the possibility of internalization to facilitate a long-lasting fluorescence signal. Ideally, this biomarker is still present on residual cells after preoperative therapy. Based on these criteria, promising clinically translatable tumor-specific biomarkers for MFS and other sarcomas have been identified in a recent systematic review [21]. Since several dedicated NIRF camera systems are already on the market, evaluating a suitable biomarker is currently one of the most important steps to orchestrating successful FGS in MFS [22]. Nevertheless, preclinical immunohistochemical (IHC) evaluation studies evaluating biomarkers for FGS in MFS are scarce and did not comprehensively compare biomarkers in tumor tissue with adjacent healthy tissue [23]. Therefore, this study aims to evaluate and compare the expression of eight previously selected promising biomarkers for FGS by IHC in MFS tissue samples with adjacent healthy tissue using an objective IHC scoring method. In addition, the effect of preoperative therapy on staining intensity is assessed to determine if this might influence the biomarker’s suitability for FGS.

## 2. Materials and Methods

### 2.1. Sample Selection and Ethical Approval

Tissue samples from MFS patients who gave informed consent for the use of their tissue at the time of surgery, were older than 18 years, and underwent surgical resection between 2008 and 2020 were eligible for this study. Slides of MFS tissue samples with tumor and adjacent healthy tissue were selected by a pathologist specializing in sarcoma. Subsequently, all available corresponding formalin-fixed paraffin-embedded (FFPE) whole tissue blocks were collected from the bone and soft tissue tumor biobank at the Leiden University Medical Center. Anonymized patient and tumor characteristics of the collected specimen were retrospectively reviewed to describe some of the patient and tumor characteristics. The study was performed with ethical approval from the institutional review board of Leiden-Den Haag-Delft (protocol number: 2020-7; METC number: B20.048) and in compliance with the Declaration of Helsinki.

### 2.2. Biomarkers

A total of eight biomarkers were selected for IHC evaluation based on their potential to detect FGS in MFS. Biomarkers were chosen based on a clinical trial that had already included MFS patients using vascular endothelial growth factor A (VEGF-A) as a target for FGS and a systematic review that selected targets for FGS in soft tissue sarcomas [19,21]. Selected biomarkers were: tumor endothelial marker-1 (TEM-1), vascular endothelial growth factor receptor-1 (VEGFR-1), vascular endothelial growth factor receptor-2 (VEGFR-2), VEGF-A, epidermal growth factor receptor (EGFR), insulin-like growth factor-1 receptor (IGF-1R), platelet-derived growth factor receptor-α (PDGFR-α), and cluster of differentiation antigen 40 (CD40, also known as TNFRS5 or p50).

### 2.3. Immunohistochemistry Procedure

The primary and secondary antibodies used for IHC evaluation were selected based on the literature and previous experience with antibodies for IHC staining in our lab [23] (Appendix A). FFPE whole tissue samples from MFS patients were collected, and 4 um-thick sections were prepared using a microtome (Leica RM2235, Leica Biosystems, Nußloch, Germany). Next, sections were deparaffinized in xylene for 15 min, rehydrated in a series of 100%, 50%, and 25% ethanol dilutions, and rinsed in demineralized water. Subsequently, endogenous peroxidase was blocked with 0.3% hydrogen peroxide in demineralized water for 20 min. Afterwards, antigen retrieval was done as described in Appendix A using the PT Link (Dako, Glostrup, Denmark), and two washing steps of 5 min in phosphate buffered saline (PBS) were performed. According to the protocol, sections were incubated in a humidified room with 150 µL primary antibodies using predetermined optimal solutions based on a series of test stainings for each biomarker on positive as well as negative controls (identified with the human protein atlas, Appendix A) [24]. Afterwards, slides were washed three times in PBS for 5 min and incubated with appropriate secondary antibodies (goat anti-mouse HRP conjugate for VEGF-A: catalog number K4001, Agilent Technologies, Santa Clara, CA, USA; and goat anti-rabbit HRP conjugate for all other biomarkers: catalog number K4003, Agilent Technologies, Santa Clara, CA, USA), followed by an additional washing step. Staining was visualized through incubation with 3,3-diaminobenzidine tetrahydrochloride solution (DAB, K3468, Agilent Technologies, Inc., Santa Clara, CA, USA) for 10 min at room temperature. Finally, sections were counterstained with Mayer’s hematoxylin solution for 30 s (Sigma-Aldrich, St. Louis, MO, USA) and rinsed. After dehydration in an incubator for 1 h at 37 °C, slides were mounted with Pertex (Leica Microsystems, Wetzlar, Germany). A negative (PBS) and conjugate control (only secondary antibody) were included to evaluate potential nonspecific staining. Slides were examined under a Zeiss AxioSkop 20 light microscope (Carl Zeiss, Jena, Germany), scanned using a PANNORAMIC 250 scanner (3DHISTECH Ltd., Budapest, Hungary), and viewed using Case Viewer (3DHISTECH Ltd., Budapest, Hungary).

### 2.4. Objective Immunohistochemistry Scoring Method

A pathologist specializing in sarcoma (who works in a tertiary referral center and has more than 10 years of experience) annotated the margin between tumor and adjacent healthy tissue in each hematoxylin and eosin (H&E)-stained MFS tissue sample using the CaseViewer application. Annotated images were saved and opened in ImageJ, an open platform for biomedical image analysis [25]. An objective and easy-to-use 5 step protocol was used to evaluate the difference in staining intensity between the tumor and adjacent healthy tissue (Appendix A). In step 1, we select the tumor and adjacent healthy tissue as two separate regions of interest (ROI), based on manual annotations from the pathologist. In step 2, the non-DAB color spectra are filtered out by using the color deconvolution option (H&E DAB) [26]. In this way, only the relevant brown DAB staining that indicates biomarker staining intensity can be assessed. In step 3, the image data is converted from the tridimensional RGB color model into monochromous grayscale (pixel intensities ranging from 0 to 255) to reflect DAB staining density. In step 4, the image is inverted, so high pixel intensities represent high DAB staining density. Step 5 is the measurements step, where we measure the mean staining intensity and standard deviation in tumor tissue and adjacent healthy tissue. These values can be used for statistical analyses to indicate the difference in staining intensity between the tumor and adjacent healthy tissue and to calculate the tumor-to-background ratio (TBR, calculated as the mean staining intensity of the tumor ROI divided by the mean staining intensity of the adjacent healthy tissue ROI), which is crucial for assessing a biomarker’s potential for FGS; a higher TBR indicates a greater potential for its use in FGS.

### 2.5. Statistical Analysis

Statistical analyses were performed using SPSS version 25 (IBM Corp., Armonk, NY, USA), and graphs were made using GraphPad PRISM (version 9, GraphPad Software, La Jolla, CA, USA). The mean and standard deviation were used because the distribution of our data was symmetric and there were no clear outliers. Descriptive statistics were used to report the TBR of each biomarker. A comparison of the mean staining intensity within one sample (tumor compared to adjacent healthy tissue) for biomarkers that showed good overall expression was performed using a paired *t*-test. A comparison of the mean staining intensity between two groups (with or without preoperative therapy) for biomarkers that showed good overall expression was performed using an unpaired *t*-test. Significance was set at a *p*-value ≤ 0.05.

## 3. Results

### 3.1. Patient Characteristics

Whole tissue blocks with vital tumor resection material and adjacent healthy tissue from seventeen MFS cases were included in this IHC evaluation study (Table 1). The total cohort had a mean age of 66 years; most of the included patients were male (*n* = 9; 53%); most tumors were located in the extremities (*n* = 14; 82%); most tumors had an intermediate or high histological grade according to the French Federation of Cancer Centers Sarcoma Group (FNCLCC) grading system (*n* = 13; 76%); most patients received preoperative therapy (*n* = 11; 65%); and the histopathological response according to the European Organization for Research and Treatment of Cancer-Soft Tissue and Bone Sarcoma Group (EORT-STBSG) was always E (≥50% stainable tumor cells; Table 1) [27,28].

### 3.2. Immunohistochemistry Results

Despite repeated scanning, annotating tumor tissue was impossible for two samples due to destructed IHC slides (loss of attachment) and bad imaging quality. We were able to include seventeen MFS cases in our IHC analyses. For the patients that had more than one FFPE sample, mean staining intensity values were averaged to ensure that each patient had one corresponding mean staining intensity value for the tumor tissue and one corresponding mean staining intensity value for the adjacent healthy tissue.

#### 3.2.1. TEM-1, PDGFR-α, and VEGF-A Are Promising Biomarkers for FGS in MFS

Diffuse, strong TEM-1 expression was observed in all MFS (100%). For all TEM-1-stained samples, there was a mean staining intensity of 81 in tumor tissue (mean standard deviation 39) and a mean TBR of 3.1. Although TEM1 staining intensity values varied greatly between different samples, expression was always higher in tumors than in adjacent healthy tissue (Figure 1). For most cases, clear differences were observed between the tumor and adjacent muscle and fat tissue (Figure 2 and Figure 3).

Diffuse, strong VEGF-A staining was also observed in all cases (100%), where the mean staining intensity of tumor tissue was 88 (mean standard deviation, 45). However, VEGF-A had a mean TBR of 2.4 because staining intensity was relatively high in adjacent healthy tissue. Moreover, in three samples, the staining intensity was higher in adjacent healthy tissue than in tumor tissue, resulting in a TBR < 1 (Figure 1).

For PDGFR-α, all MFS tumor cases (100%) also showed diffuse, strong expression, with a mean staining intensity of 64 (mean standard deviation, 36) and a mean TBR of 1.8. Similar to the VEGF-A staining, three samples had a higher staining intensity in adjacent healthy tissue compared to tumor tissue, resulting in a TBR < 1 (Figure 1).

#### 3.2.2. VEGFR-1, VEGFR-2, EGFR, IGF-1R, and CD40 Did Not Show Good Overall Expression

Unfortunately, VEGFR-1, VEGFR-2, EGFR, IGF-1R, and CD40 were not expressed in 13/17 (76%) MFS cases (Appendix A). Based on the four (24%) cases that showed biomarker expression, VEGFR-1 had a mean tumor staining intensity of 21, with a mean TBR of 1.5. In the same cases, the mean tumor staining intensity was 25 for VEGFR-2 (mean TBR 0.7), 7 for EGFR (mean TBR 0.9), 7 for IGF-1R (mean TBR 0.5), and 18 for CD40 (mean TBR 2).

### 3.3. Preoperative Therapy Did Not Significantly Affect TEM-1, PDGFR-α, and VEGF-A Expression

Six patients (35%) from the MFS tumor group did not receive any preoperative therapy. Subgroup analysis did not show statistically significant differences between mean tumor intensity and TBR values of TEM-1, VEGF-A, and PDGFR-α when comparing groups of stained tissue samples with or without preoperative therapy (Table 2). Surprisingly, mean tumor intensity scores were higher in the subgroup that received preoperative therapy compared to the group without preoperative therapy.

## 4. Discussion

FGS could improve adequate tumor resections by delineating tumor tissue from adjacent healthy tissue. This IHC study evaluated eight candidate biomarkers for FGS in seventeen MFS tumors in comparison with adjacent healthy tissue. A pathologist specializing in sarcomas annotated tumor tissue, and staining intensity was assessed using an objective scoring method. All MFS tissue samples showed expression of TEM-1, PDGFR-α, and VEGF-A. However, the mean TBR was higher for TEM-1 than for VEGF-A and PDGFR-α. Therefore, this study identified TEM-1 as the most suitable biomarker for FGS in MFS. Interestingly, no statistically significant differences were observed between the intensity scores of TEM-1, PDGFR-α, and VEGF-A with or without preoperative therapy, suggesting FGS based on these markers might still be feasible after preoperative therapy. Although IHC results might not directly correspond to clinical results, our findings align with the previous study published by de Gooyer et al. [23]. Here, 34 MFS tissue microarrays showed high expression for VEGF-A and TEM-1 and moderate expression for PDGFR-α. Additional H&E and TEM-1 stainings were performed on ten FFPE blocks with tumor and adjacent healthy tissue. Although results were not objectively assessed, clear tumor- to-adjacent healthy tissue borders were reported in all TEM-1-stained sections. The added value of our study is the comprehensive and objective assessment of eight previously selected promising biomarkers for FGS in a higher amount (seventeen) of MFS tissue samples with adjacent healthy tissue. For each biomarker, this study described the number of MFS tissue samples that displayed expression, the mean tumor staining intensity, and the mean TBR. This broader approach gives a better understanding of which biomarker is most suitable for FGS in MFS. Steinkamp et al. could delineate several STS subtypes, including seven MFS, with Bevacizumab-800CW targeting VEGF-A in vivo [19]. In this paper, fluorescence TBRs of 2.0–2.5 were reported with doses of 10–25 mg tracer. Although overexpression of VEGF-A in STS has been described in several studies, the tumor-specificity of VEGF-A in MFS has not been reported [29,30,31]. Our study demonstrated three cases where VEGF-A was expressed more in adjacent healthy tissue compared to tumor tissue, which would be dramatic in the case of VEGF-A-based FGS as it could lead to over-resection with more wound complications and increased functional impairments as a consequence. Based on our IHC evaluation, this phenomenon seems less expected for TEM-1 because it has always been expressed more in tumor tissue compared to adjacent healthy tissue. Therefore, a TEM-1 targeting moiety, such as a humanized, clinically approved monoclonal antibody like Ontuxizumab or an anti-TEM-1 antibody fragment (1C1m) conjugated to a fluorophore, could be a more specific alternative than Bevacizumab-800CW [32,33]. This study has several strong points. The first strength is the inclusion of seventeen whole tissue samples with MFS tumor and adjacent healthy tissue assessed by a pathologist specializing in sarcoma. Second, the selected biomarkers in this IHC evaluation study had a scientifically robust basis because they were selected as promising biomarkers for FGS by a clinical trial that had already included MFS patients and a previous systematic review that selected targets for FGS in soft tissue sarcomas [19,21]. Another positive aspect, which should be the basis of IHC research with FFPE material, is the methodological transparency: optimal primary antibody solutions were predetermined based on a series of test stainings for each biomarker on positive as well as negative controls that were identified with the human protein atlas (Appendix A) [24]. Ambiguous staining results on control tissue were repeated with different antibody concentrations and sometimes on other positive or negative control tissues. Comparing IHC studies for the purpose of FGS should normally be done with caution due to the variability of results depending on the type of antibodies, dilutions, epitopes, clones, or staining protocols used [34]. However, by using the five step protocol (Appendix A), we were able to objectively quantify the intensity of each biomarker’s staining, and our results depended less on subjective assessment by pathologists with interobserver variability [35]. Biomarker staining densities are still somewhat affected by pathology scanner characteristics and settings, color deconvolution algorithms, and variability in manual annotations. Negative aspects of the current objective scoring method are the lack of comparison with a manual scoring method, the lack of correction for cell density, and the fact that parts without tissue, such as empty fat vacuoles, were also part of the intensity measurements. Yet, we believe this method should be adopted by others because it is unbiased, reproducible, and reliable in assessing the difference in a biomarker’s staining intensity between tumor and adjacent healthy tissue for its potential in FGS. Limitations of the study include the lack of use of standard, semi-automated image annotation software to score the stained tissue. Automated image annotation algorithms, such as those that can be trained in QuPath, could also provide a more objective evaluation of biomarker expression [36]. Although QuPath has been previously used in our group, we decided not to do so because our cohort is relatively small, which complicates validation algorithms [37]. In addition, QuPath could also wrongly classify out-of-focus tissue areas and staining artifacts. Another limitation is the absence of diagnostic biopsy material from MFS patients to directly compare the preoperative therapy effect on biomarker expression. Response to neoadjuvant therapy could lead to decreased biomarker expression intensity if there is a large percentage of non-stainable tumor cells. All patients in our cohort had ≥50% stainable tumor cells (according to the EORTC-STBSG), which could explain why preoperative therapy did not significantly affect TEM-1, PDGFR-α, and VEGF-A expression [28]. From other types of cancer, such as breast and pancreatic cancer, we know that IHC staining patterns change after neoadjuvant therapy, but for TEM-1 in MFS, this remains largely unknown [38,39]. Although our analysis of the preoperative therapy effect did not have sufficient statistical power to draw conclusions, de Gooyer et al. also state that preoperative radiotherapy did not significantly influence TEM-1 expression in MFS tissue [23]. This observation could also be due to an increased expression with enhanced tumor grade and is particularly relevant as preoperative radiotherapy is currently advised for most intermediate- and high-grade MFS patients (21% of MFS patients in the Netherlands according to a recent epidemiological study) [5,14,30,40]. As briefly highlighted before, TEM-1 targeted tracer based on Ontuxizumab or 1C1m and conjugated to a fluorophore should be further investigated. For clinical translation, it is highly important that a targeted tracer against a promising biomarker (i.e., TEM-1 targeted tracer for MFS) is applicable to the majority of patients. Although our sample size is too small per subgroup to draw hard conclusions, we did not notice any association between staining intensity and tumor characteristics, like grade or location. Ideally, one universal FGS tracer should be implemented to improve clinical outcomes for all oncological patients. However, current literature provides mixed results concerning TEM-1 expression in other cancer subtypes. On one hand, TEM-1 is predominantly expressed by fibroblasts and a subset of pericytes associated with tumor vessels but not by tumor endothelium in melanoma, ovarian, lung, and brain cancer. On the other hand, several studies conclude that TEM-1 is strongly expressed on tumor cells, tumor vasculature, and stroma in the majority of soft tissue and bone sarcomas [41,42,43,44,45,46,47]. Future studies are needed to determine if TEM-1 is indeed expressed in other sarcomas and if TEM-1-specific tracers are suitable for FGS. For now, the main benefit of a TEM-1 targeted tracer seems to reside in delineating MFS from adjacent healthy tissue, which could theoretically lead to reduced LR and improved survival rates.

## 5. Conclusions

Fluorescence-guided surgery (FGS) has the potential to improve the surgical outcomes of myxofibrosarcoma (MFS) patients by delineating tumors from adjacent healthy tissue. This study evaluated eight biomarkers for their potential in FGS in seventeen MFS samples with adjacent healthy tissue. Expression of tumor endothelial marker-1 (TEM-1), vascular endothelial growth factor-A (VEGF-A), and platelet-derived growth factor receptor α (PDGFRα) was observed in all MFS tumors. However, TEM-1 was identified as the most promising biomarker for FGS in MFS because it had the highest tumor-to-background intensity ratio regardless of preoperative therapy. Therefore, TEM-1-targeted FGS tracers should be further investigated to optimize MFS treatment.

## Figures and Tables

**Figure 1 biomedicines-11-00982-f001:**
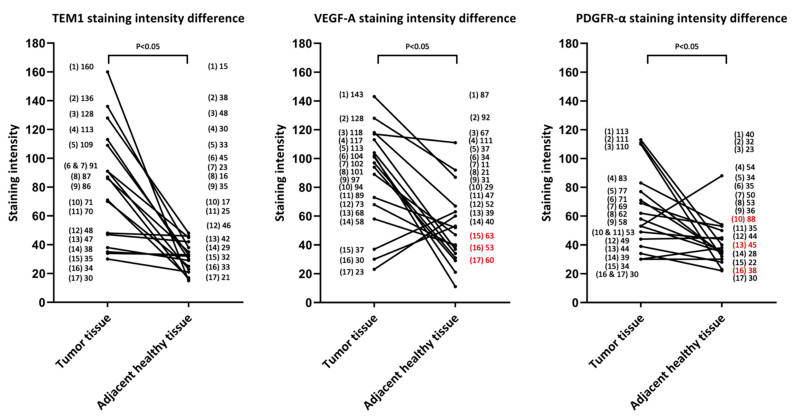
Tissue intensity difference between tumor and adjacent healthy tissue per sample. Exact mean staining intensity values per case are displayed on the left- and right-hand y-axes. TEM-1 displays the highest difference in tumor- versus-adjacent healthy tissue staining intensity. For three cases, VEGF-A staining intensity and PDGFR-α staining intensity were higher in adjacent healthy tissue compared to tumor tissue (shown in red). Samples had a statistically significant higher staining intensity in tumors compared to adjacent healthy tissue for the TEM-1, VEGF-A, and PDGFR-α staining groups (*p* < 0.05).

**Figure 2 biomedicines-11-00982-f002:**
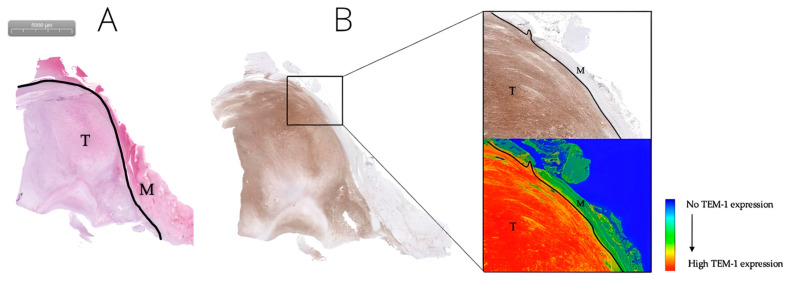
Grade 2 MFS located in the upper arm of a 63-year-old female. H&E staining of MFS tumor tissue (T), the tumor border (black line), and adjacent muscle tissue (M) (**A**). Corresponding TEM-1 staining (**B**): the upper right 10× enlarged image displays tumor tissue (T), the tumor border (black line), and adjacent healthy muscle tissue (M). The lower-right corresponding image shows a gradient map of the same TEM-1 staining, which clearly delineates tumor from adjacent healthy tissue.

**Figure 3 biomedicines-11-00982-f003:**
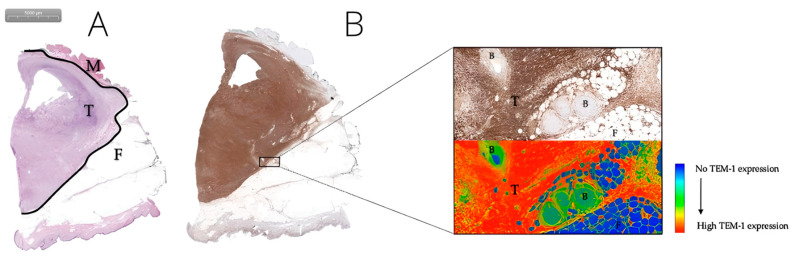
A grade 3 MFS tumor located in the upper arm of a 64-year-old female. H&E staining of MFS tumor tissue (T), the tumor border (black line), and adjacent muscle and fat tissue (M and F) (**A**). Corresponding TEM-1 staining (**B**): the upper right 30× enlarged image displays tumor tissue (T), blood vessels (B), and adjacent fat tissue (F). The lower-right corresponding image shows a gradient map of the same TEM-1 staining, which clearly delineates tumor from adjacent healthy tissue.

**Table 1 biomedicines-11-00982-t001:** Patient and tumor characteristics.

Patient	Gender	Age	Tumor Location	Histological Grade (FNCLCC)	Preoperative Therapy	Response Score (EORTC-STBSG)
1	Female	77	Upper extremity	2	None	Not applicable
2	Male	56	Upper extremity	1	None	Not applicable
3	Female	62	Lower extremity	1	Radiotherapy	E
4	Male	68	Trunk	1	None	Not applicable
5	Male	55	Lower extremity	2	None	Not applicable
6	Female	81	Trunk	3	None	Not applicable
7	Female	63	Upper extremity	2	None	Not applicable
8	Male	47	Lower extremity	1	Radiotherapy	E
9	Male	64	Upper extremity	2	Radiotherapy	E
10	Female	64	Upper extremity	3	Radiotherapy	E
11	Male	78	Lower extremity	2	Radiotherapy	E
12	Male	49	Lower extremity	2	Radiotherapy	E
13	Female	70	Lower extremity	2	Radiotherapy	E
14	Male	70	Trunk	2	Radiotherapy	E
15	Male	79	Lower extremity	3	Chemotherapy	E
16	Female	67	Upper extremity	2	Radiotherapy	E
17	Female	66	Upper extremity	2	Radiotherapy	E

Abbreviations: FNCLCC = French Federation of Cancer Centers Sarcoma Group classification system; 1 = low grade; 2 = intermediate grade; and 3 = high grade. EORTC-STBSG = European Organization for Research and Treatment of Cancer-Soft Tissue and Bone Sarcoma Group; A = no stainable tumor cells; B = single stainable tumor cells or small clusters (overall below 1% of the whole specimen); C = ≥1%–<10% stainable tumor cells; D = ≥10%–<50% stainable tumor cells; and E = ≥50% stainable tumor cells.

**Table 2 biomedicines-11-00982-t002:** Intensity scores and tumor-to-background ratios of TEM-1, PDGFR-α, and VEGF-A categorized by preoperative therapy.

	With Preoperative Therapy, *n* = 11, All with Response Score E (EORTC-STBSG)	Without Preoperative Therapy, *n* = 6	*p*-Value
TEM-1			
Mean tumor intensity	85	74	0.60
Mean TBR	3.5	2.2	0.29
VEGF-A			
Mean tumor intensity	96	73	0.38
Mean TBR	2.7	1.8	0.19
PDGFR-α			
Mean tumor intensity	70	52	0.21
Mean TBR	1.9	1.5	0.41

Abbreviations: EORTC-STBSG = European Organization for Research and Treatment of Cancer-Soft Tissue and Bone Sarcoma Group; A = no stainable tumor cells; B = single stainable tumor cells or small clusters (overall below 1% of the whole specimen); C = ≥1%–<10% stainable tumor cells; D = ≥10%–<50% stainable tumor cells; and E = ≥50% stainable tumor cells. TBR = tumor-to-background ratio (calculated as the mean staining intensity of the tumor divided by the mean staining intensity of the adjacent healthy tissue).

## Data Availability

The data presented in this study are available on request from the corresponding author.

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
