# Peer review of "Immunohistochemical Evaluation of Candidate Biomarkers for Fluorescence-Guided Surgery of Myxofibrosarcoma Using an Objective Scoring Method"

_biomedicines, 2023, doi:10.3390/biomedicines11030982_

Round 1
Reviewer 1 Report
This is an intersting immunohistochemical study on myxofibrosarcomas evaluating potential biomarkers for FGS.
There are some points:
There are some discrepancies between the complexity of the introduction and discussion and the essential result that TEM-1 is the most promising marker as already described by de Gooyer et al. I would recommend to focus more precisely and to shorten the manuscript by doing so.
The messages are in some ways to black and white and relativizationarguments can be helpful.
Some of the mentioned strong arguments for this study (discussion) are actually the base of research immunohistochemical studies on FFPE material.
Whether the computer based scoring system is optimal seems to be questionable as the authors did not compare their results with manual scoring possibly preventing staining artefacts.
When you talk about therapy response you could discuss the histomorphology as replacement of reactive tissue has influence on the staing pattern.
Minor points:
Please check your terminology: tumorsample vs patient (e.g. immunohistochemical results).
Reviewer 2 Report
Thank you that you allowed me to review this paper.
The paper is well written, with a good transition from one section to the next. Each chapter is well-organized and well-presented in a clear manner. The introduction effectively sets the stage for the rest of the paper, and found the arguments to be well-supported by relevant literature.
Figures are well presented, but classical H/E stains of the tumors compared with immunohistochemistry are recommended to assess if TEM-1 is useful in everyday practice.
The discussion section provides an insightful analysis of the results and their implications, and the conclusion summarizes the main findings and provides a clear summary of the paper’s contribution to the field.
In conclusion, I believe that this paper is a valuable contribution and with some minor revisions, I would be happy to recommend it for publication.
Reviewer 3 Report
The manuscript entitled “Immunohistochemical Evaluation of Candidate Biomarkers for Fluorescence-Guided Surgery of Myxofibrosarcoma Using an Objective Scoring Method” by Rijs Z et al. describes the identification of promising tumor-specific biomarkers to delineate the tumor tissue during surgery.
The authors have presented an interesting topic in the field of myxofibrosarcoma and the manuscript looks like well written and organized. The paper should be considered after major revisions.
1. The study population is quite small and should be better described in the first results section. I suggest the authors to increase the casistic;
2. The authors should provide more detailed information about patient’s characteristics, such as the presence of metastasis, local recurrence, follow up and other;
3. Another limitation is the poor description of the molecular status of the specimens. The authors should provide information about the positivity to common diagnostic markers, such as S100, SMA, CD34 or MDM2;
4. The authors should discuss if there are significantly correlation between the IHC results and patient’s characteristics, such as tumor locations or grading;
5. The authors should describe the different therapeutic strategies for patients affected by MFS to better understand the importance of FGS. The references entitled” Myxofibrosarcoma landscape: diagnostic pitfalls, clinical management and future perspectives” doi: 10.1177/17588359221093973 and “A Functional Precision Oncology Approach to Identify Treatment Strategies for Myxofibrosarcoma Patients” doi: 10.1158/1541-7786.MCR-21-0255 should be added to the manuscript.
Round 2
Reviewer 3 Report
The manuscript is now acceptable to be published in "Biomedicines".